# Dual-comb photothermal spectroscopy

Qiang Wang [1,2,6✉], Zhen Wang [3,6✉], Hui Zhang[1,2], Shoulin Jiang[4], Yingying Wang[5], Wei Jin[4] & Wei Ren [3✉]

Dual-comb spectroscopy (DCS) has revolutionized optical spectroscopy by providing broadband spectral measurements with unprecedented resolution and fast response. Photothermal spectroscopy (PTS) with a pump-probe configuration offers a highly sensitive gas sensing method, which is normally performed using a single-wavelength pump laser. The merging of PTS with DCS may enable a spectroscopic method by taking advantage of both technologies, which has never been studied yet. Here, we report dual-comb photothermal spectroscopy (DC-PTS) by passing dual combs and a probe laser through a gas-filled anti-resonant hollow-core fiber, where the generated multi-heterodyne modulation of the refractive index is sensitively detected by an in-line interferometer. As an example, we have measured photothermal spectra of acetylene over 1 THz, showing a good agreement with the spectral database. Our proposed DC-PTS provides opportunities for broadband gas sensing with super-fine resolution and high sensitivity, as well as with a small sample volume and compact configuration.

---

[1] State Key Laboratory of Applied Optics, Changchun Institute of Optics, Fine Mechanics and Physics, Chinese Academy of Sciences, 130033 Changchun, China. [2] University of Chinese Academy of Sciences, 100049 Beijing, China. [3] Department of Mechanical and Automation Engineering, and Shenzhen Research Institute, The Chinese University of Hong Kong, New Territories, Hong Kong SAR, China. [4] Department of Electrical Engineering, The Hong Kong Polytechnic University, Kowloon, Hong Kong SAR, China. [5] Institute of Photonics Technology, Jinan University, 510632 Guangzhou, China. [6]These authors contributed equally: Qiang Wang, Zhen Wang. ✉email: wangqiang@ciomp.ac.cn; wangzhen@link.cuhk.edu.hk; renwei@mae.cuhk.edu.hk

Optical frequency combs (OFCs) have significantly promoted the development of many research fields in the past decades such as time and frequency metrology[1,2], attosecond science[3], optical communication[4], and precision spectroscopy[5,6]. This light source presents an optical spectrum composed of a set of evenly spaced and phase-coherent laser lines. The combination of broad spectral bandwidth and high spectral resolution makes OFCs particularly suitable for high-precision molecular spectroscopy and multi-species gas detection. Among different OFC-based spectroscopic methods[7–12], dual-comb spectroscopy (DCS) provides a versatile approach to Fourier transform interferometry without any moving parts, leading to many advantages such as high resolution, short acquisition time, high signal-to-noise ratio (SNR), and free from size constraints.

DCS is widely studied for gas absorption spectroscopy that a high-bandwidth photodetector is used to measure the comb transmission through the gas medium. This technique requires a long mutual coherence time for averaging and a long light-gas interaction path to enhance the detection sensitivity. The mutual coherence time between two combs has approached 1860 s with the feed-forward locking technique[13]. Although DCS has been demonstrated with different kinds of comb sources, most studies used a short single-pass gas cell with high-concentration gas samples just for demonstration purposes[14–18]. It is possible to increase the effective path length using a multipass gas cell or an optical cavity[12,19]. However, the multipass cell requires large mirrors and has a bulky size. The optical loss per reflection also limits the use of low-power OFCs[20]. In comparison, an optical cavity can increase the optical path by orders of magnitude with less optical loss. Cavity-enhanced dual-comb spectroscopy has been reported with ultra-high sensitivity[12,21,22], which requires complex optical setup and feedback control electronics to achieve the so-called two-point locking[11]. The spectral bandwidth may also be constrained by the dispersion of the cavity mirrors. Hence, new strategies are demanded for DCS that can achieve broadband, high-resolution and ultrasensitive detection in a compact system. Photoacoustic spectroscopy (PAS), an indirect absorption detection method, has been recently explored by several research groups using OFCs[23–26]. Based on a Fourier transform spectrometer with a comb source, researchers measured broadband photoacoustic spectra of methane using a micro-cantilever[23,24]. Very recently, dual-comb PAS has been demonstrated for measuring acetylene[25] and polymer films[26].

Interestingly, photothermal spectroscopy (PTS) is another important method widely used for gas sensing[27–30]. In PTS, the modulated light absorption generates periodic heating of a gas sample, resulting in the modulation of gas refractive index (RI). As the RI modulation is proportional to the laser intensity, it is promising to achieve ultra-high sensitivity by use of a hollow-core fiber (HCF) with a small mode field diameter (MFD). For instance, by using a hollow-core photonic bandgap fiber with a MFD of 11 µm, an optical intensity of ~40 kWcm$^{-2}$ can be obtained for an input power of 15 mW[27]. A pump-probe configuration is often adopted for PTS, where the pump laser-induced RI modulation is sensitively detected by an interferometer with a probe laser. Since the first adoption of HCFs in PTS with a Mach-Zehnder interferometer[27], different types of RI detection schemes have been reported including heterodyne interferometer[28], Fabry-Pérot interferometer[29], and mode-phase-difference method[30]. Additionally, when the pump laser wavelength is tuned away from the absorption line, or the gas sample is non-absorbing (i.e., zero gas like nitrogen), PTS has zero output if the system is properly designed, and thus behaves as a background-free detection method[29,31]. Rather than performing photothermal detection with a single-wavelength pump laser, PTS combined with the dual-comb source has the potential to achieve a compact gas sensor with background-free, ultra-high sensitivity, and broadband detection, which has never been realized previously.

In this work, a technique of dual-comb photothermal spectroscopy (DC-PTS) is reported. We use an electro-optic dual-comb source to generate the photothermal RI modulation, which is sensitively detected by an in-line Fabry-Pérot interferometer (FPI). Such a highly versatile all-fiber sensing configuration is described in detail. Our experimental results reveal the current DC-PTS can provide tooth-resolved photothermal measurements of the $(\nu_1 + \nu_3)$ band of acetylene ($C_2H_2$), spanning over 1 THz. With an average optical power of 15 mW for the dual-comb source, we achieve a minimum detection limit of 8.7 ppm $C_2H_2$ over the coherent averaging time of 1000 s. To our knowledge, this is the first demonstration of broadband photothermal detection using frequency combs.

## Results

**Theory.** Figure 1 presents the concept of the proposed DC-PTS. We use dual electro-optic frequency combs as the pump source for the proof-of-concept demonstration. The two combs with a

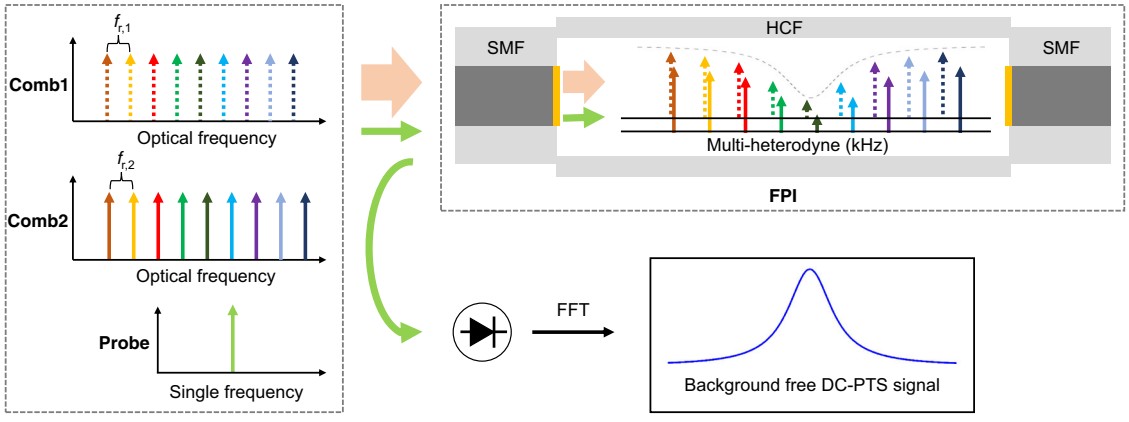

**Fig. 1 Principle of dual-comb photothermal spectroscopy (DC-PTS).** FPI Fabry-Pérot interferometer, SMF single-mode fiber, HCF hollow-core fiber, $f_{r,1}$, $f_{r,2}$ repetition frequencies of the two frequency combs, FFT Fast Fourier Transform. Two frequency combs heterodyne against each other to provide a set of amplitude modulations at the down-converted evenly-spaced frequencies. The combs and a probe laser are coupled into a FPI formed by a HCF sandwiched by two solid-core SMFs. The refractive index of the target gas in the HCF is modulated by the pump comb source, leading to the phase modulation of the probe laser. The output of the probe laser from the FPI is used to retrieve the multi-heterodyne photothermal signal by Fourier transform.

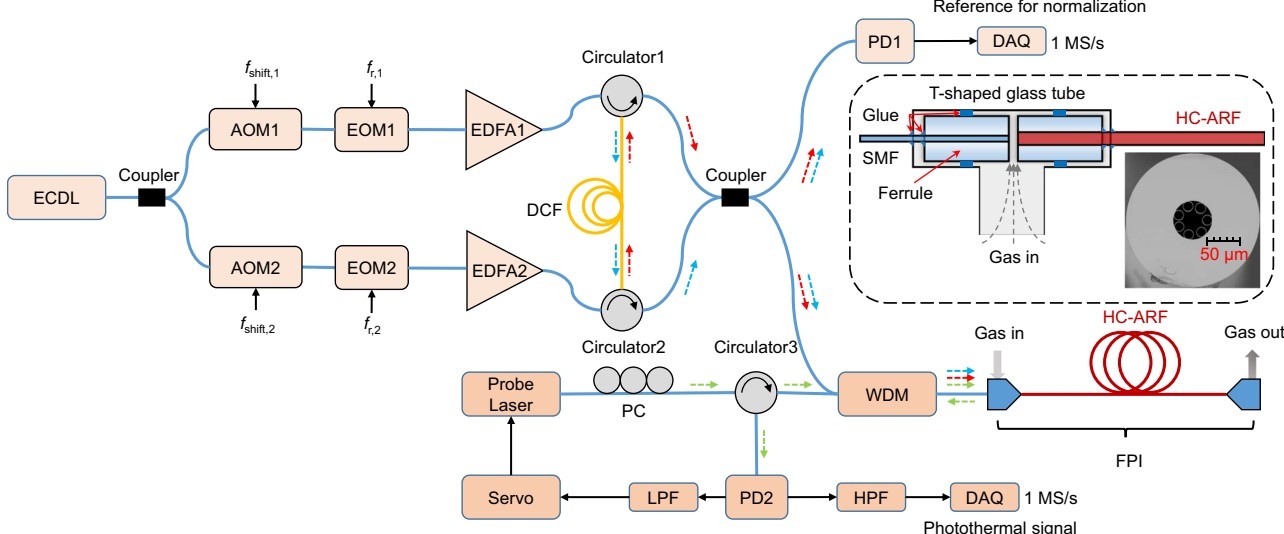

**Fig. 2 Schematic of the DC-PTS setup.** ECDL external cavity diode laser, AOM acousto-optic modulator, EOM electro-optic modulator, EDFA erbium-doped optical fiber amplifier, DCF dispersion compensation fiber, PD photodetector, WDM wavelength division multiplexer, PC polarization controller, FPI Fabry-Pérot interferometer, LPF low-pass filter, HPF high-pass filter, DAQ data acquisition with 16-bit vertical resolution, SMF single-mode fiber. Inset: scanning electron microscope (SEM) image of the cross-section of the hollow-core anti-resonant fiber (HC-ARF), and the schematic interface between HC-ARF and SMF. The detailed structure of the FPI is described in Methods section. The pump combs and the probe laser are delivered into the FPI via a WDM. The FPI output of the probe laser carrying the phase modulation information is picked up by the WDM and a circulator (Circulator3). The low-frequency component of the photodetector signal (PD2) is used to lock the probe wavelength to the quadrature point of the interference fringe via a laser servo to obtain the largest frequency-to-amplitude conversion factor (see Supplementary Fig. S2). The PC is used to optimize the contrast of interference fringes. The high-pass filtered time-domain photothermal interferograms are digitized continuously at a sampling rate of 1 MS/s and directly Fourier transformed to retrieve the spectrum. Spectral average in the frequency domain is applied to improve the SNR. Note that the DAQ card and all the radio-frequency synthesizers are referenced to a rubidium clock.

set of discrete evenly-spaced and phase-coherent optical frequency components, $f_{n,1}$ and $f_{n,2}$, can be expressed by

$$f_{n,1} = n \cdot f_{r,1} + f_0 + f_{shift,1} \qquad (1)$$

$$f_{n,2} = n \cdot f_{r,2} + f_0 + f_{shift,2} \qquad (2)$$

where $n$ (0, ±1, ±2, etc.) is the comb line index; $f_{r,j}$ and $f_{shift,j}$ ($j = 1$ and 2) are the repetition rate and optical frequency shift, respectively; $f_0$ is the optical carrier frequency which is identical for the two combs when they are generated from the same source.

In DCS, one frequency comb heterodynes against the other to down-convert the optical frequencies into the radio frequency (RF) spectrum, which can be detected by a high-bandwidth photodetector. The difference of the two repetition rates and the difference of the optical frequency shifts determine the RF comb spacing and the center frequency of the RF spectrum, respectively. In this study, we tune the down-converted RF combs to a scope of tens of kHz so that the photothermal effect of gas-phase analyte can be efficiently excited after absorbing the comb light. The multi-heterodyne signal composes of distinguishable beatnotes between pairs of comb teeth at the radio frequency $\nu_n$

$$\nu_n = |f_{n,1} - f_{n,2}| = n \cdot \Delta f_r + \Delta f_{shift} \qquad (3)$$

where $\Delta f_r = f_{r,1} - f_{r,2}$ and $\Delta f_{shift} = f_{shift,1} - f_{shift,2}$. Hence, the $n$th comb line is modulated at the frequency $\nu_n$, acting as the $n$th amplitude-modulated pump laser. If this comb line is absorbed by the gas medium, the gas density (or refractive index) is modulated at the respective radio frequency $\nu_n$. As a result, a probe laser that is made collinear with the pump combs experiences a phase modulation at $\nu_n$. When the comb lines that can be absorbed by the gas molecules pass through the gas medium, the complex phase modulation signal can be detected by an optical

interferometer and Fourier transformed to a photothermal spectrum.

As introduced previously, it is quite attractive to perform PTS in a HCF due to its highly efficient light-gas interaction and the significantly increased light intensity within the hollow core of a μm-sized MFD. For gas absorption by the $n$th pair of comb teeth, the induced phase modulation for the probe laser along the HCF can be expressed by Jin et al.[27]

$$\Delta\phi_n = k^* A L \overline{\chi} \overline{P}_{pump,n} \qquad (4)$$

where $k^*$ is a coefficient inversely proportional to the cross-section area of the HCF mode field; $A$ is the peak absorption coefficient; $L$ is the length of the HCF; $\overline{\chi}$ is the normalized line-shape function of the absorption feature; $\overline{P}_{pump,n}$ is the square root of a product of the optical power for the $n$th pair of comb teeth propagating through the HCF.

**Experimental setup.** Figure 2 depicts the experimental setup of DC-PTS based on an all-fiber electro-optic dual-comb source. The two combs generated from the same continuous wave (CW) external-cavity diode laser (ECDL) are designed with repetition rates of 500 MHz and 500.0001 MHz, and optical frequency shifts of 25 MHz and 25.03 MHz, respectively (see Methods section). The dual-comb light is divided into two branches, one of which is used as the photothermal pump. A portion of the comb light is monitored by a photodetector to provide a synchronous reference for normalizing the possible non-uniform comb intensity distribution over a broad bandwidth (see Methods section). The FPI used in this work consists of a 1572-nm probe laser, and a hollow-core anti-resonant fiber (HC-ARF) sandwiched by two single-mode fibers (SMFs). The HC-ARF also acts as a gas cell with a length of 7 cm, an air core diameter of 28 μm and a diameter of holey region of 56 μm, providing a gas sampling

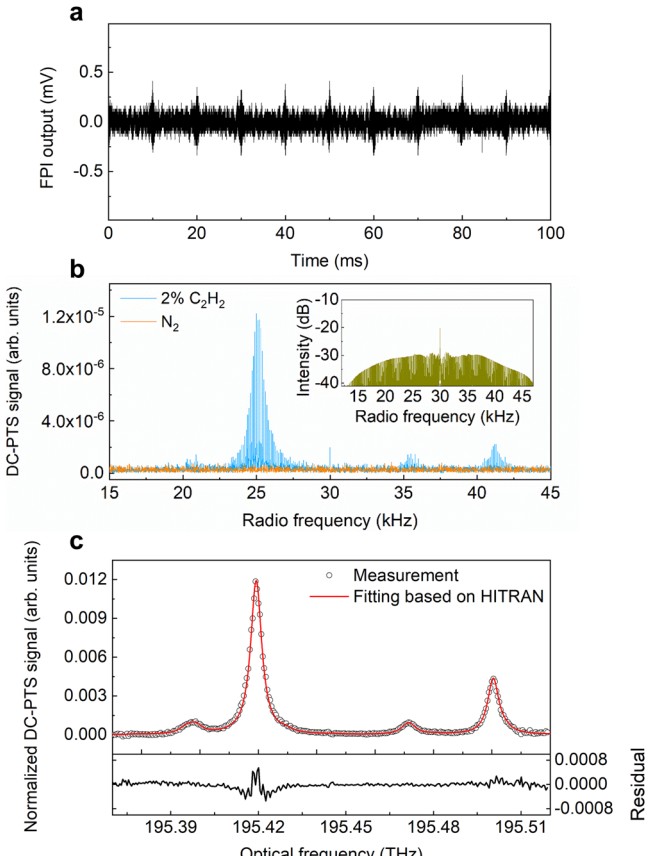

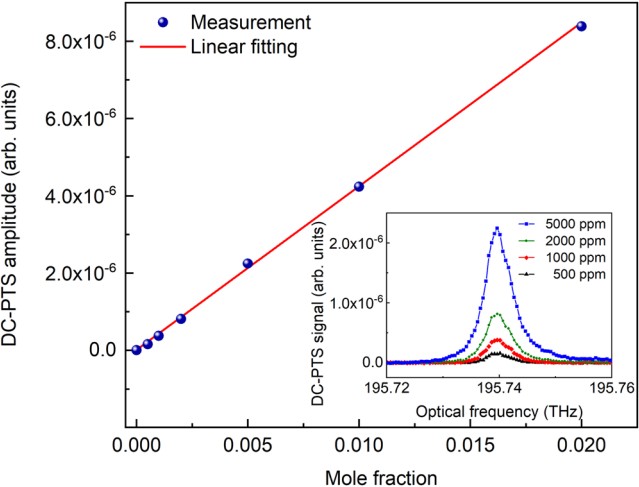

**Fig. 3 Representative DC-PTS signal of $C_2H_2$ measured at the central frequency of 195.444 THz. a** Time-domain photothermal interferograms recorded by the FPI. **b** Fourier-transformed photothermal multi-heterodyne signal of 2% $C_2H_2$ and pure $N_2$ for a measurement time of 100 ms. The inset shows the corresponding dual-comb reference spectrum recorded by the photodetector PD1. **c** Normalized dual-comb photothermal spectrum of $C_2H_2$ for a measurement time of 10 s. The residual between the measurement and spectral fitting is plotted at the bottom panel.

**Fig. 4 Variation of the DC-PTS amplitude with gas concentration.** The DC-PTS system shows an excellent linear response with an $R^2$ value of 0.999. Inset: representative DC-PTS spectra of $C_2H_2$ with different concentrations.

volume of only 0.17 μL. The densely packed beatnotes centered at 30 kHz have a frequency span from 15 kHz to 45 kHz and a line spacing of 100 Hz, over which the FPI has a relatively high and flat-top response (see Supplementary Fig. S1).

We choose $C_2H_2$ as an example to demonstrate the DC-PTS concept. Firstly, a gas mixture of 2% $C_2H_2/N_2$ is introduced to the HC-ARF and the optical carrier frequency of the comb source is tuned to 195.444 THz. All the measurements are conducted at 1 bar and 296 K. Figure 3a shows the representative time-domain interferograms. The bursts reproduce with a periodicity of 10 ms, corresponding to the repetition frequency difference of the dual-comb source. Figure 3b shows the Fourier-transformed photo-thermal signal of $C_2H_2$, showing a comb-line-resolved spectrum over 15–45 kHz. One can easily distinguish two strong absorption lines P(15) and P(14) centered at ~25 kHz (195.42 THz in optical domain) and ~41 kHz (195.50 THz in optical domain), respectively; as well as two relatively weak lines at ~21 kHz (195.40 THz) and ~35.4 kHz (195.47 THz), respectively. It is of interest to observe one single comb line at 30 kHz, which is located at the far wing of the P(15) line. This comb line derives from the beatnote of the residual carrier CW lasers, leading to a much higher intensity than any other bilateral beatnotes. Such an influence can be mitigated by normalizing the direct photo-thermal measurement with a reference multi-heterodyne dual-comb signal, or by employing a pair of electro-optic modulators

(EOMs) with higher extinction ratios. We choose the former method in this study due to its simpler implementation. The inset graph of Fig. 3b presents the multi-heterodyne dual-comb spectrum that is simultaneously recorded by the photodetector (PD1 in Fig. 2). Dividing the direct DC-PTS signal by the reference multi-heterodyne dual-comb spectrum, we can also mitigate the influence of the non-uniform intensity distribution of the comb source over broadband. In order to obtain the absorption features based on the HITRAN database, Fig. 3c depicts the normalized $C_2H_2$ spectrum for a measurement time of 10 s, which is in good agreement ($R^2$ value of 0.998) with the absorption spectral fitting using the HITRAN database[32]. The relative residual around the absorption peak at 195.42 THz is below 5%, which may be caused by the spectral fitting uncertainty. The Voigt profile commonly used for line-shape calculations neglects higher order effects such as line mixing and speed dependence[33]. The possible drift of the ECDL wavelength, which induces a frequency shift of comb teeth in each beatnote, may also cause spectral distortion during the averaging process. The fringe noise observed in the residual is believed to be caused by the optical interference effect of the FPI (see Supplementary Fig. S3).

**Linear response and broadband detection.** We then evaluate the sensor response by measuring $C_2H_2/N_2$ mixtures with different concentrations. A stronger absorption line P(11) is targeted by tuning the central frequency to 195.764 THz. A dual-comb power of 15 mW is delivered to the HC-ARF in this test. Different gas mixtures are generated by diluting the certified 2% $C_2H_2/N_2$ with $N_2$ using a gas dilution system. The variation of the amplitude of the DC-PTS signal with $C_2H_2$ concentration is plotted in Fig. 4, showing an excellent linear relationship ($R^2$ value of 0.999).

By tuning the ECDL frequency in a step-wise manner and stitching 15 sequential spectra, the current electro-optical dual-comb spans more than 1 THz. For the gas sample of 2% $C_2H_2/N_2$, each DC-PTS measurement with normalization is conducted for a measurement time of 100 s, and then the spectral stitching is applied to obtain the entire spectra shown in Fig. 5. The rovibrational lines of the $(\nu_1 + \nu_3)$ band of $C_2H_2$ are well-resolved in this spectral range, showing a good agreement with the simulated spectrum using the HITRAN database[32].

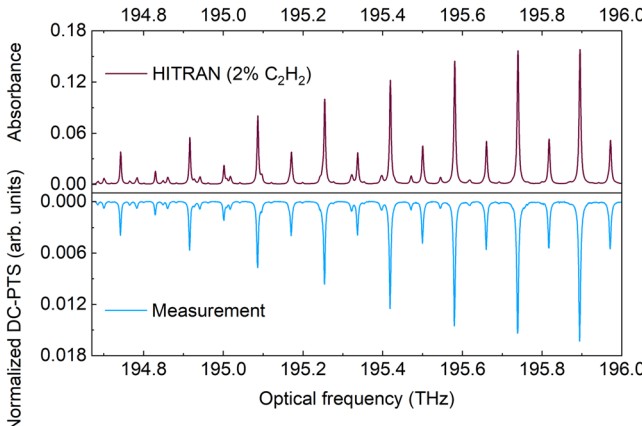

**Fig. 5 Comparison of the broadband DC-PTS measurement with the spectral simulation based on the HITRAN database.** The gas mixture of 2% $C_2H_2$ is used in this test. The multi-heterodyne photothermal signal is normalized to account for the non-uniform comb intensity distribution and FPI response over broadband.

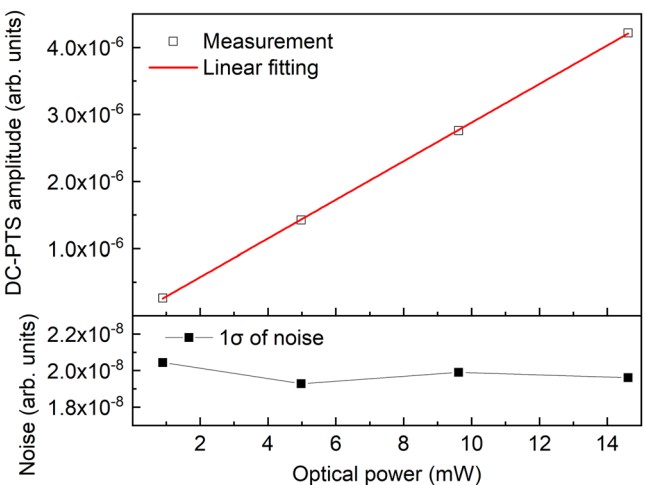

**Fig. 6 Variation of the amplitude of DC-PTS signal of 1% $C_2H_2$ and the noise (1σ) with the comb power.** The P(11) line is analyzed for a measurement time of 10 s.

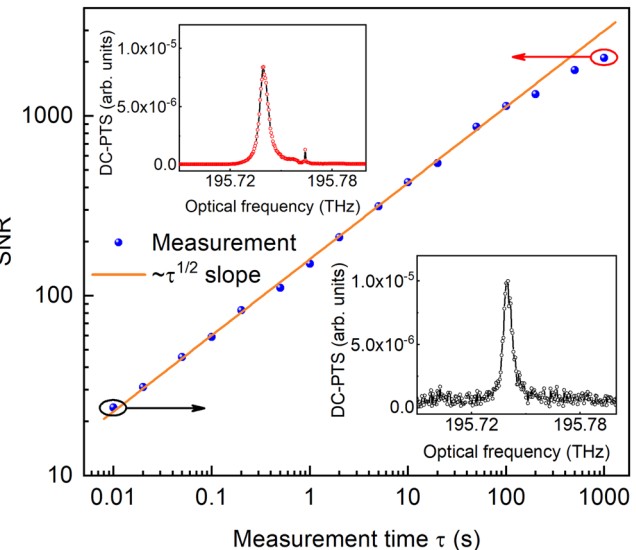

**Fig. 7 Variation of the SNR of 2% $C_2H_2$ with the measurement time.** The measurement SNR is defined as the ratio of the peak magnitude of the PTS response to the standard deviation of noise in the non-absorption range. Insets: typical DC-PTS spectra measured at 10 ms and 1000 s, respectively.

## Discussion

The reported results reveal that the DC-PTS system provides a way of dual-comb spectroscopic measurements. It is of interest to discuss the possible methods of obtaining a higher SNR so that the DC-PTS can become more feasible for trace gas detection.

Firstly, the dependence of the DC-PTS signal on the pump power is studied. Here the pump power is measured at the coupler output of the two combined combs. By adjusting the pump power between 1 mW and 15 mW using a fiber-optic attenuator, Fig. 6 plots the corresponding amplitude of the DC-PTS signal of 1% $C_2H_2/N_2$ and the noise level measured for 10 s. The amplitude increases linearly with power, while the noise remains almost unchanged. The commercial EDFA with a higher gain has achieved an output power of 1.5 W for gas sensing[34]. Considering the comb power of 15 mW used in this work, we expect to increase the sensitivity by two orders of magnitude.

We further investigate the SNR by varying the measurement time. The definition of SNR can be found in Methods section. Figure 7 illustrates that the SNR increases linearly with the square root of time from 10 ms up to 100 s, and further increases until the maximum recording duration of 1000 s with a slight deviation from the ideal trend line. Such a slight deviation may derive from the phase noise of combs caused by low-frequency temperature variation and mechanical vibration. For a mixture of 2% $C_2H_2/N_2$, the current DC-PTS system achieves a SNR of ~1100 at the measurement time of 100 s, and ~2300 at 1000 s. Hence, a minimum detection limit (MDL, 1σ) of 8.7 ppm is determined for the dual-comb power of 15 mW, the fiber length of 7 cm, and the measurement time of 1000 s (see Methods section). It should be noted that such a long-term averaging method may be not suitable for applications that require a fast time response.

Finally, we mention several other methods that can further enhance the system performance. For the electro-optic combs used in this work, it is feasible to change the comb spacing by tuning the repetition rate of the electrical pulse used for driving EOMs. We expect a repetition frequency of up to 5 GHz, which is mainly limited by the bandwidth of the pulse generator. A higher SNR can be obtained by increasing the repetition frequency as the power of each comb tooth rises with the increased line spacing, given a constant average power. However, we must consider a tradeoff between the measurement SNR and spectral resolution in actual spectroscopic measurements. Additionally, we use an HC-ARF with a length of only 7 cm in this work. As the phase modulation can be accumulated with the fiber length shown in Eq. (4), it is possible to increase the SNR by simply using a longer HCF. It has been recently reported that silica HC-ARFs have achieved a record low transmission loss of 0.28 dB/km in the C and L bands[35]. Hence, we expect to enhance the dual-comb photothermal signal by another two orders of magnitude if a 10-m-long HC-ARF is used for the FPI, considering its negligible loss of pump comb power. Note that the optical interference effect of the FPI may cause fluctuations of the PTS signal. Although the amplitude variation in the current study is < 5% (see Supplementary Fig. S3), it could become serious when the FPI is exposed to external disturbance such as temperature variation and mechanical vibration. This issue can be possibly mitigated by applying a broadband anti-reflection (AR) coating to the two fiber facets over the wavelength range of comb light so that a FPI is not formed for the pump. Meanwhile, the probe wavelength can be selected beyond the spectral range of the AR coating so that the reflections for the probe are still relatively high to form a FPI with

a reasonable interference fringe contrast for the detection of photothermal phase (or RI) modulation.

In summary, we report the concept and proof-of-principle demonstration of DC-PTS using an electro-optic dual-comb combined with an in-line FPI. Because of the inherently high mutual coherence of this dual-comb source and the exquisite frequency agility, the dual-comb generates the multi-heterodyne modulation of the RI of the gas sample in the HC-ARF. A moderate optical span of hundreds of GHz is mapped to the frequency response range of the FPI. The photothermal spectra of $C_2H_2$ pumped by the dual-comb source can be obtained by the Fourier transform of the interferograms generated from the FPI within a short measurement time, i.e., as short as 10 ms in this work. The spectra of $C_2H_2$ over 1 THz are stitched by simply tuning the optical frequency of the ECDL. We achieve an MDL of 8.7 ppm for the dual-comb power of 15 mW and fiber length of 7 cm. Several strategies for further improving the SNR are discussed. Hence, our study opens a opportunity to realize broadband, high-resolution and sensitive gas detection. Although demonstrated for electro-optic combs, we believe it is quite promising to apply our technique for mid-infrared combs generated from micro-resonators[36], quantum cascade lasers[37], and interband cascade lasers[38] with extra merits of stronger gas absorption and reduced size for integration.

## Methods

**Electro-optic dual-comb source.** For the generation of electro-optic dual-comb, a CW-ECDL is divided into two branches via a 50:50 fiber coupler and then each branch is connected in parallel to an acousto-optic modulator (AOM), which shifts the frequency of the CW laser by 25 MHz and 25.03 MHz, respectively. These two frequency-shifted optical beams are intensity modulated by two EOMs, driven by 50-ps pulses at slightly different repetition frequencies of 500 MHz and 500.0001 MHz, respectively. The electrical pulses have peak-to-peak amplitudes of about 6.7 V. The generated optical pulse trains are amplified by EDFAs and subsequently counter launched into a single 1-km nonlinear optical fiber with high normal dispersion (dispersion of $-130$ ps nm$^{-1}$ km$^{-1}$, and low dispersion slope of $-0.15$ ps nm$^{-2}$ km$^{-1}$) for spectral broadening. The broadened combs are separated by a pair of circulators (Cirlulator1 and Circulator2 shown in Fig. 2) at each end of the nonlinear fiber. The dual combs are mixed and split into two beams by a coupler for photothermal pumping and normalization purposes, respectively.

**In-line Fabry-Pérot interferometer.** By manually aligning a HCF to an input SMF pigtail and an output SMF pigtail, a FPI can be formed by natural reflections at the two joints between the HCF and SMFs, with a total insertion loss of 1.2 dB. The relative position between facets of the two kinds of fibers is finely adjusted to achieve a maximum contrast of interference for the probe laser. For gas inlet and outlet, both joints have a gap of 1–2 μm and are sealed within a T-shaped glass tube (inset graph of Fig. 2). The HCF also acts as a gas cell in the interferometer. We use a HC-ARF with a length of 7 cm instead of the commercial hollow-core photonic band-gap fiber (HC-PBF). Such a new fiber exhibits a relatively larger air core (28 μm) to facilitate the gas loading and the total gas consumption is only 0.17 μL. The mode purity of the HC-ARF is much better and mode interference noise is essentially avoided by careful alignment of the input SMF and the HC-ARF.

**Photothermal spectrum normalization.** The direct DC-PTS signal is shown in Fig. 3b. However, the photothermal effect is proportional to the pump comb power shown in Fig. 6. Thus, the original DC-PTS signal is convolved by the effects of the non-flat FPI response (see Supplementary Fig. S1) and the non-uniform power distribution of comb lines. Based on the tooth-to-tooth mapping relationship, the direct DC-PTS signal is normalized to the intensity of the dual-comb spectrum and the response of the FPI.

**Definition of SNR and MDL.** The SNR is defined as the peak value of the retrieved P(11) line divided by the standard deviation of the signal in the range of $195.77 - 195.80$ THz, where the absorption of $C_2H_2$ is minimal. The radio-frequency scale is converted to an optical scale based on the frequency compression factor of $f_r/\Delta f_r = 5 \times 10^6$, as well as the knowledge of the optical carrier frequency. The MDL is obtained by evaluating the SNR of the measured DC-PTS signal for the $C_2H_2$ sample with a calibrated concentration.

**Reporting summary**. Further information on research design is available in the Nature Research Reporting Summary linked to this article.

## Data availability

The data that support the plots within this paper and other findings of this study are available on Zenodo (https://doi.org/10.5281/zenodo.6362609). All other data used in this study are available from the corresponding authors upon reasonable request.

## Code availability

No original codes have been developed for this article. Whereas a simple Matlab program used to compute the Fourier transforms is available from the corresponding authors upon reasonable request.

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

## Acknowledgements

This research was supported by the National Natural Science Foundation of China (NSFC) (62005267, Q.W.; 51776179, W.R.), Strategic Priority Research Program of Chinese Academy of Sciences (XDA17040513, XDA22020502, Q.W.), General Research Fund (14209220, W.R.) from the University Grants Committee, Innovation and Technology Fund (MHP/049/19, GHP/123/19GD, W.R.) from the Innovation and Technology Commission, Scientific Instrument Developing Project of the Chinese Academy of Sciences (YJKYYQ20190037, Q.W.), the Second Comprehensive Scientific Investigation of the Qinghai-Tibet Plateau (2019QZKK020802, Q.W.).

## Author contributions

Q.W., Z.W., and W.R. conceived the idea, designed the experiments, discussed the results and prepared the manuscript. Q.W. built the systems and conducted the experiments. Z.W. assisted in building the systems and conducting the experiments. H.Z. completed the spectral analysis. S.J. and W.J. fabricated the fiber-based interferometer. Y.W. provided the hollow core anti-resonant fiber. Q.W. and W.R. supervised and coordinated the project.

## Competing interests

The authors declare no competing interests.
