## [Peer Review file · Nature Communications]

REVIEWER COMMENTS

Reviewer #1 (Remarks to the Author):

The paper demonstrates the new principle of absorption measurements using the dual-comb technique and photothermal measurements. The demonstration is clear and shows interesting features. The subject is entirely in the journal's scope; the paper is free from basic errors and is suitable for publishing.

However, the authors sometimes forget their own goal seeking - to demonstrate the principle and the usefulness of the combination and make some too positive remarks about the suitability of the technique for chemical analysis, like in the final part "Hence, our study opens a new opportunity to realize broadband, high-resolution and ultrasensitive chemical analysis." (lines 269-271). In fact, it is not proven as the chemical analysis in this paper is absent; there is only proof of concept with the model mixture in ideal conditions. After a lot of work, it could be implemented as a method of chemical analysis, but this is not the case in this paper. Therefore, I suggest rewriting the discussion seriously.

My concerns can be summed up as follows:

1. Line 66. "Additionally, photothermal spectroscopy is a background-free method for chemical analysis". First, not all the methods of photothermal spectroscopy are the same; please use the names of the exact technique here and in all the text.

Second, without an explanation, this statement is utterly wrong. By its nature, from the viewpoint of chemical analysis, photothermal spectroscopy is NOT a background-free method. For many years, the work of many people has been directed towards chemical and physical sample preparation or data handling to deal with background problems. This paper does not show that this technique will be background-free from this point of view. Therefore, this statement should be fully explained or removed.

2. The limit of detection under the conditions is an Instrument detection limit (IDL) - without chemical factors. I suggest changing the abbreviation from MDL to IDL not to confuse the applied audience, suggesting that the authors provide method detection limit with the contribution of sampling, sample preparation, etc.

3. Also, I do not understand the choice of the 1σ criterion even for IDL; this should be explained explicitly. Otherwise, it seems that the authors try to show the record-breaking sensitivity for its own sake, thus canceling their own intention to offer the chemical-analysis possibilities, not the physical principle alone.

4. The sensitivity of the proposed technique is not unique. When the authors claim "ultrasensitive chemical analysis", they should at least that the practical goal: what samples, what sensitivity, and under what conditions should be measured, and if this technique has no competitor techniques like gas sensors, detectors, optoacoustic measurements, etc. The reasons should be given and discussed. Right now, the text says nothing from the view of applied chemists, not taking into account the instrument and analysis cost, and it is an example of 'yet another sensitive tool'. Thus, once again, either this whole discussion should be removed or seriously decreased as this is not proved, or otherwise, it should be expanded to all these questions.

5. The selection of topical papers on photothermal spectroscopy "Interestingly, photothermal spectroscopy (PTS) is another important method widely used in spectroscopy and microscopy [27–30]." is strange. This claim is too broad, without specific tasks, samples, techniques, or even the sample state: (solid, gas, or liquid). It has no reviews, no papers on chemical analysis, and paper 30 is essential but outdated, and the authors have much more relevant articles right now. I suggest expanding this list, showing more discussion, sticking to gas analysis, and discussing these papers in the discussion section.

Reviewer #2 (Remarks to the Author):

The authors presented a photothermal spectroscopy scheme based on dual EO optical frequency combs. To some extent, this could be a modified alternative approach to the photoacoustic dual-comb spectroscopy demonstration in Ref. 25. With the adoption of a piece of hollow-core fiber as the gas sample holder and the all-optical FP interferometry technique to pick up signals, this new scheme could offer advantages and flexibility not yet available to the previous ones. There are additional justification or clarification to further support the claimed superiority of the proposed method over others, before acceptance for publication.

1) What is the optical filtering effect of the FPI setup they used? The transmission spectrum should be provided. How would that affect the dual-comb spectrum and their results? There are some discussions about further increasing the HCF length to a much larger one, and that would make things much trickier. In view of this unavoidable spectral modulation by FPI, additional discussion of the potential limitations would be helpful.

2) The authors should have provided more technical details of their setup, which makes it somewhat difficult to judge some of the claims in their discussions, such as the generated comb spectrum (is it the

same under different powers), the driving power of the comb generator, the power levels at different points in the setup and the insertion optical loss of the FPI. Also, for the very long data acquisition case (like 1000s), are the data continuously acquired? How are they processed? by directly doing FFT? It's not clearly described.

3) It is claimed that the spectroscopic results can be obtained over a ~ 1 THz span, and considering the center wavelength of the EO dual combs can be easily tuned, what is the limiting factor that prevented further increasing this measurement range to cover other absorption peaks?

4) There are some discussions about obtaining improved SNR by increasing the pump power to Watt level. How would that impact the FPI setup when there would be non-negligible optical loss of the incident power that may be transferred to thermal instability or other side effects of this fiber-optic approach? Is it really feasible to extrapolate the current results at moderate pump power to a much larger one?

5) Additional discussion and quantitative comparison with similar schemes like the photoacoustic dual-comb one would be helpful. Does this photothermal scheme offer advantages considering the SNR under similar optical power and other conditions?

6) It was said that the fibers were spliced together but there is a small gap between the fibers to inject gas into the HCF. How was this implemented? Does that affect the stability and reliability of this setup?

7) The residual shown in Fig.3 is relatively large at the main absorption peak, indicating the existence of some systematic errors. There should be more discussions about that. Also, the definition of SNR shown in the method section seems to only focus on the random noise away from the larger residual region and may overestimate the quality of the results.

Reviewer #3 (Remarks to the Author):

The paper presents a demonstration of dual-comb photothermal spectroscopy using a cw laser as a probe and a hollow core fiber as a gas cell. While dual-comb photoacoustic spectroscopy has been previously demonstrated, to my knowledge, this is the first demonstration of dual-comb photothermal spectroscopy. The paper was well written and provided a good overview of technique. As such, I believe that this will be potentially useful to other researchers. I do however have a few questions/comments that would be nice to be addressed before publication.

1. How is the data collection/averaging accomplished? Is the DAQ clock linked to the repetition rate? Do you stream the entire 100 or 1000 second time series to disk? Is averaging done in the spectral domain or time domain?

2. For the data in Figure 5, how many sequential scans were used to cover the full range?

3. The authors showed the linear response of the DC-PTS amplitude with optical power. This should correspond to increasing SNR, but that was not shown. Could the authors show an SNR metric (or give numbers in the text)?

4. The schematic of the profile of the HC-ARF is a bit confusing to interpret. I think this is because the rods are the same color as the surrounding hollow areas. Maybe the rods could be made with a cross-hatch pattern to distinguish them better?

5. The gas inlet and outlet method was a bit confusing to me. It sounds like the SMF is spliced to the HCF, but that a cap is left between the SMF and HCF. How is this achieved?

6. One final point is that it would be interesting to have a comparison of the SNR of the photothermal system compared to a linear absorption measurement. This can be done by estimating the SNR of the linear measurement using the published absorbance noise from other systems as well as the HITRAN calculated absorbance spectrum. Is the maximum achievable SNR (assuming higher laser power for photothermal) higher for photothermal? If not, then what would be the expected advantage of photothermal spectroscopy?

Manuscript number: NCOMMS-21-46571

Title: Dual-comb Photothermal Spectroscopy

Correspondence Authors: Prof. Qiang Wang, Dr. Zhen Wang, and Prof. Wei Ren

Response to Reviewers

We appreciate all the reviewers' comments and suggestions that help to improve our manuscript and strengthen the impact. We provide here our point-by-point response and the edits made in the revised manuscript.

Reviewers' comments are in black, responses are in blue.

Reviewer #1:

The paper demonstrates the new principle of absorption measurements using the dual-comb technique and photothermal measurements. The demonstration is clear and shows interesting features. The subject is entirely in the journal's scope; the paper is free from basic errors and is suitable for publishing.

However, the authors sometimes forget their own goal seeking - to demonstrate the principle and the usefulness of the combination and make some too positive remarks about the suitability of the technique for chemical analysis, like in the final part "Hence, our study opens a new opportunity to realize broadband, high-resolution and ultrasensitive chemical analysis." (lines 269-271). In fact, it is not proven as the chemical analysis in this paper is absent; there is only proof of concept with the model mixture in ideal conditions. After a lot of work, it could be implemented as a method of chemical analysis, but this is not the case in this paper. Therefore, I suggest rewriting the discussion seriously.

Reply: We agree with the reviewer that the main purpose of this paper is to report the concept of DC-PTS and the proof-of-principle demonstration of spectroscopic measurement of gases. After a lot of follow-up research and systematic validation, we expect this method can be used by applied chemists for chemical analysis. To avoid the possible overstatement of the research goal, we have replaced 'chemical analysis' by 'gas sensing' and revised the discussion in the manuscript.

My concerns can be summed up as follows:

1. Line 66. "Additionally, photothermal spectroscopy is a background-free method for chemical analysis". First, not all the methods of photothermal spectroscopy are the same; please use the names of the exact technique here and in all the text.

Second, without an explanation, this statement is utterly wrong. By its nature, from the viewpoint of chemical analysis, photothermal spectroscopy is NOT a background-free method. For many years, the work of many people has been directed towards chemical and physical sample preparation or data handling to deal with background problems. This paper does not show that this technique will be background-free from this point of view. Therefore, this statement should be fully explained or removed.

Reply: The exact technique we used in this work is photothermal spectroscopy with the pump-probe configuration and FPI. More description of photothermal spectroscopy and the state-of-the-art is given in the Introduction of the revised manuscript.

In photothermal spectroscopy and other spectroscopic methods for gas detection, the background-free is defined as, if the pump laser wavelength is tuned away from the absorption line of the target gas, or the gas for analysis does not absorb the light (i.e., zero gas like nitrogen), the system will output zero signal. It is in contrast to the traditional laser absorption spectroscopy (some people also call it transmission spectroscopy) in which an absorption event results in a reduction of the transmitted light signal from a large background. Such a particular definition of 'background-free' is used in the Introduction section. Previous studies of photothermal gas sensors with the pump-probe configuration have proved this feature [1-2]. However, a background may exist for PTS if the interferometer picks up unwanted interferences from the environment. We revised the Introduction by adding further discussion and clarification.

However, the term 'background-free' is possibly defined in a different way from the view of applied chemists. For instance, there may be numerous unknown species in the mixture that also absorb the light and cause the background signal. Thus, chemical and physical sample preparation or sophisticated data analysis is needed to deal with the so-called background problem. Hence, the issue raised by the reviewer is mainly due to the different understanding of the definition of 'background-free', either from a spectroscopic view or from a view of chemical analysis. We believe the introduction of PTS for gas detection and its characteristics are clear now with the added clarification.

[1] *Waclawek, J. P., Bauer, V. C., Moser, H., & Lendl, B. 2f-wavelength modulation Fabry-Perot photothermal interferometry. Opt. Express* **24**, 28958-28967 (2016).

[2] *Yao, C., Gao, S., Wang, Y., Wang, P., Jin, W., & Ren, W. MIR-pump NIR-probe fiber-optic photothermal spectroscopy with background-free first harmonic detection. IEEE Sens. J.* **20**, 12709-12715 (2020).

2. The limit of detection under the conditions is an Instrument detection limit (IDL) - without chemical factors. I suggest changing the abbreviation from MDL to IDL not to confuse the applied audience, suggesting that the authors provide method detection limit with the contribution of sampling, sample preparation, etc.

Reply: The minimum detection limit (MDL, 1σ), in the original manuscript, is widely used in gas sensing applications [3-5]. Its definition is provided in the Methods of the revised manuscript. We believe it is reasonable to keep the term MDL and limit our discussion to gas sensing.

[3] *Wu, H., Dong, L., Zheng, H., Yu, Y., Ma, W., Zhang, L., Yin, W., Xiao, L., Jia, S., & Tittel, F. K. Beat frequency quartz-enhanced photoacoustic spectroscopy for fast and calibration-free continuous trace-gas monitoring, Nat. Commun.* **8**, 15331 (2017).

[4] *Wei, T., Zifarelli, A., Dello Russo, S., Wu, H., Menduni, G., Patimisco, P., Sampaolo, A., Spagnolo, V., & Dong, L. High and flat spectral responsivity of quartz tuning fork used as infrared photodetector in tunable diode laser spectroscopy. Appl. Phys. Rev.* **8**, 041409 (2021).

[5] *Waclawek, J. P., Moser, H., & Lendl, B. Balanced-detection interferometric cavity-assisted photothermal spectroscopy employing an all-fiber-coupled probe laser configuration. Opt. Express* **29**, 7794-7808 (2021).

3. Also, I do not understand the choice of the 1sigma criterion even for IDL; this should be explained explicitly. Otherwise, it seems that the authors try to show the record-breaking sensitivity for its own sake, thus canceling their own intention to offer the chemical-analysis possibilities, not the physical principle alone.

Reply: The MDL is obtained by evaluating the SNR of the DC-PTS signal of the concentration calibrated C₂H₂ sample. The definition of SNR and MDL are added to the Methods section.

4. The sensitivity of the proposed technique is not unique. When the authors claim "ultrasensitive chemical analysis", they should at least that the practical goal: what samples, what sensitivity, and under what conditions should be measured, and if this technique has no competitor techniques like gas sensors, detectors, optoacoustic measurements, etc. The reasons should be given and discussed. Right now, the text says nothing from the view of applied chemists, not taking into account the instrument and analysis cost, and it is an example of 'yet another sensitive tool'. Thus, once again, either this whole discussion should be removed or seriously decreased as this is not proved, or otherwise, it should be expanded to all these questions.

Reply: As this work focuses on the introduction of a novel concept, we agree with the reviewer and have confined our discussion to spectroscopic analysis of gases rather than chemical analysis throughout the manuscript.

5. The selection of topical papers on photothermal spectroscopy "Interestingly, photothermal spectroscopy (PTS) is another important method widely used in spectroscopy and microscopy [27–30]." is strange. This claim is too broad, without specific tasks, samples, techniques, or even the sample state: (solid, gas, or liquid). It has no reviews, no papers on chemical analysis, and paper 30 is essential but outdated, and the authors have much more relevant articles right now. I suggest expanding this list, showing more discussion, sticking to gas analysis, and discussing these papers in the discussion section.

Reply: After the revision of the manuscript based on the reviewer's comments, we make the paper focused on the spectroscopic analysis of gases or gas sensing. The Introduction section focuses on PTS and its state-of-the-art, which is revised and updated by more focused references. Discussion on the technical progress has been added to the revised manuscript.

Reviewer #2:

The authors presented a photothermal spectroscopy scheme based on dual EO optical frequency combs. To some extent, this could be a modified alternative approach to the photoacoustic dual-comb spectroscopy demonstration in Ref. 25. With the adoption of a piece of hollow-core fiber as the gas sample holder and the all-optical FP interferometry technique to pick up signals, this new scheme could offer advantages and flexibility not yet available to the previous ones. There are additional justification or clarification to further support the claimed superiority of the proposed method over others, before acceptance for publication.

1) What is the optical filtering effect of the FPI setup they used? The transmission spectrum should be provided. How would that affect the dual-comb spectrum and their results? There are some discussions about further increasing the HCF length to a much larger one, and that would make things much trickier. In view of this unavoidable spectral modulation by FPI, additional discussion of the potential limitations would be helpful.

Reply: The FPI has the optical filtering effect as commented by the reviewer. The dual-comb spectrum can be affected by the FPI due to the slight reflections of the comb light at the two fiber facets forming the FPI. The optical spectra of the dual-comb before and

after passing through the FPI are provided in the Supplementary. By dividing these two spectral data, we can observe the FPI filtering effect caused by the optical interference of the comb light. The interference may introduce additional noise if the FPI experiences external disturbances such as temperature variation and mechanical vibration. When a longer HCF is applied, the interference fringes become denser due to the reduced free spectral range of the FPI. However, thanks to the low reflectivity of the FPI for the pump comb, the interference contrast is low in the current demonstration, which does not affect the pump power significantly. This effect can be mitigated by coating the two fiber facets of the FPI to minimize reflections for the pump comb. We have added more discussion in the revised manuscript.

2) The authors should have provided more technical details of their setup, which makes it somewhat difficult to judge some of the claims in their discussions, such as the generated comb spectrum (is it the same under different powers), the driving power of the comb generator, the power levels at different points in the setup and the insertion optical loss of the FPI. Also, for the very long data acquisition case (like 1000s), are the data continuously acquired? How are they processed? by directly doing FFT? It's not clearly described.

Reply: The dual-comb spectrum remained unchanged under different power levels in this study as the optical power was adjusted using a fiber attenuator after the dual-comb source. For the comb generation, the RF driving signal applied to the two EOMs has a pulse width of 50 ps and a pulse amplitude of about 6.7 V. The average optical power is about 100 mW before being launched into the dispersion compensation fiber. The maximum dual-comb power is about 15 mW before entering the FPI. The total insertion loss of the FPI is about 1.2 dB, each joint with a loss of 0.6 dB. As the sampling rate is 1 MS/s, the data can be acquired and streamed to computer hard disk continuously in 1000 s using a data acquisition card and a LabVIEW program. The acquired raw data are directly Fourier transformed and averaged in the frequency domain. The above technical details have been added to the revised manuscript.

3) It is claimed that the spectroscopic results can be obtained over a ~1 THz span, and considering the center wavelength of the EO dual combs can be easily tuned, what is the limiting factor that prevented further increasing this measurement range to cover other absorption peaks?

Reply: The spectrum span of our current system is limited by the two EDFAs available in our lab.

4) There are some discussions about obtaining improved SNR by increasing the pump power to Watt level. How would that impact the FPI setup when there would be non-negligible optical loss of the incident power that may be transferred to thermal instability or other side effects of this fiber-optic approach? Is it really feasible to extrapolate the current results at moderate pump power to a much larger one?

Reply: We believe it is feasible to increase the pump power to Watt level for the following reasons. The insertion loss of each fiber joint is about 0.6 dB, corresponding to 13% loss (includes 4% interface reflection). Hence, the dissipation loss is about 9%, most of which will escape from the joint. Only very little can be absorbed to generate a thermal effect considering the low absorption of SMFs within the EDFA spectral range. In an AR-HCF, more than 99% of pump power is in the hollow core. The power handling ability of both SMF and AR-HCF is considerably greater than 1 W. The damage threshold of cleaved SMF ends is much greater than 1 W. Additionally, a recent study of long-term stability (4 hours) at high pump power (300 mW) has shown little thermal instability [1].

[1] Liu, F., Bao, H., Ho, H. L., Jin, W., Gao, S., & Wang, Y. Multicomponent trace gas detection with hollow-core fiber photothermal interferometry and time-division multiplexing. *Opt. Express*, **29**, 43445-43453 (2021).

5) Additional discussion and quantitative comparison with similar schemes like the photoacoustic dual-comb one would be helpful. Does this photothermal scheme offer advantages considering the SNR under similar optical power and other conditions?

Reply: We compare the performance between dual-comb photoacoustic spectroscopy in reference 25 and the current DC-PTS in terms of detection limit in the table below:

	Power	Averaging time	Detection limit
This work	15 mW	1000 s	8.7 ppm
DCPAS	20 mW	1000 s	10 ppm

Considering the absorption lines in both works have the similar line-intensity, the current study shows a slightly better sensitivity. Besides, the DC-PTS proposed in this study enables an all-fiber configuration with an extremely small gas sample volume of 0.17 μL . It should be noted that dual-comb photoacoustic spectroscopy in ref. 25 relies on the detection of the comb-induced acoustic wave by a microphone, while dual-comb photothermal spectroscopy in this study reports the concept of measuring comb-induced variation of refractive index by an in-fiber interferometer. Considering the quite different working principles between these two methods, we believe it is not suitable to compare the sensitivity of the two works based on the proof-of-principle results. The objective of both studies is to demonstrate a new concept to the spectroscopic and gas sensing communities. The SNR of both methods can be further improved after numerous follow-up studies.

6) It was said that the fibers were spliced together but there is a small gap between the fibers to inject gas into the HCF. How was this implemented? Does that affect the stability and reliability of this setup?

Reply: Instead of the traditional fusion splicing of optical fibers, we align the fibers manually to generate a small gap between them. Then all the fibers are fixed using UV epoxy glue to ensure stability and reliability. An inset graph has been added to Fig. 2 in the revised manuscript for clarification. Besides, we have replaced 'mechanically splicing' by 'manually aligning' in the Methods of the revised manuscript.

7) The residual shown in Fig. 3 is relatively large at the main absorption peak, indicating the existence of some systematic errors. There should be more discussions about that. Also, the definition of SNR shown in the method section seems to only focus on the random noise away from the larger residual region and may overestimate the quality of the results.

Reply: The relative residual around the absorption peak at 195.42 THz is below 5 %, which may be caused by the spectral fitting uncertainty. The Voigt profile commonly used for line-shape calculations neglects higher order effects such as line mixing and speed dependence [2]. The possible drift of the ECDL wavelength, which induces a frequency shift of comb teeth in each beatnote, may also cause spectral distortion during the averaging process. In addition, the fringe noise in the residual is believed to be caused by the optical filtering effect of the FPI (see response to Question 1 and the added discussion in Supplementary). We have added more discussion about the residual in the revised manuscript.

Regarding the definition of SNR, in DCS and other absorption spectroscopy techniques, both the random noise and fitting residual influence the SNR because the signal is derived from the baseline normalization and fitting process. In PAS or PTS, however, the spectrum is derived with no need for baseline. Thus, the SNR is normally defined as the peak value of signal divided by the noise under the non-absorption condition [3-5]. We adopt the same definition in this work which is described in the Methods section. The dual-comb spectrum in Fig. 3(c) with power normalization is only used for comparison purposes with the HITRAN database.

[2] Rieker, G. B., Giorgetta, F. R., Swann, W. C., Kofler, J., Zolot, A. M., Sinclair, L. C., ... & Newbury, N. R. *Frequency-comb-based remote sensing of greenhouse gases over kilometer air paths*. *Optica* **1**, 290-298 (2014).

[3] Wu, H., Dong, L., Zheng, H., Yu, Y., Ma, W., Zhang, L., Yin, W., Xiao, L., Jia, S., & Tittel, F. K. *Beat frequency quartz-enhanced photoacoustic spectroscopy for fast and calibration-free continuous trace-gas monitoring*, *Nat. Commun.* **8**, 15331 (2017).

[4] Jin, W., Cao, Y., Yang, F. & Ho, H. L. *Ultra-sensitive all-fibre photothermal spectroscopy with large dynamic range*. *Nat. Commun.* **6**, 7767 (2015).

[5] Waclawek, J. P., Moser, H., & Lendl, B. *Balanced-detection interferometric cavity-assisted photothermal spectroscopy employing an all-fiber-coupled probe laser configuration*. *Opt. Express* **29**, 7794-7808 (2021).

Reviewer #3:

The paper presents a demonstration of dual-comb photothermal spectroscopy using a cw laser as a probe and a hollow core fiber as a gas cell. While dual-comb photoacoustic spectroscopy has been previously demonstrated, to my knowledge, this is the first demonstration of dual-comb photothermal spectroscopy. The paper was well written and provided a good overview of technique. As such, I believe that this will be potentially useful to other researchers. I do however have a few questions/comments that would be nice to be addressed before publication.

1. How is the data collection/averaging accomplished? Is the DAQ clock linked to the repetition rate? Do you stream the entire 100 or 1000 second time series to disk? Is averaging done in the spectral domain or time domain?

Reply: As the sampling rate is 1 MS/s, the data is acquired and streamed to the computer hard disk continuously in 1000 s using a data acquisition card and a LabVIEW program. The acquired raw data are Fourier transformed and then averaged in the frequency domain. During the data acquisition, the DAQ card and all the other radio-frequency synthesizers are referenced to a rubidium clock. We have added the information in the revised manuscript.

2. For the data in Figure 5, how many sequential scans were used to cover the full range?

Reply: The measured DC-PTS signal in Fig. 5 spans over 1 THz, which is stitched from 15 sequential scans by tuning the ECDL frequency in a step-wise manner. We have added the information in the revised manuscript.

3. The authors showed the linear response of the DC-PTS amplitude with optical power. This should correspond to increasing SNR, but that was not shown. Could the authors show an SNR metric (or give numbers in the text)?

Reply: We added the noise level at different power levels in Fig. 6 of the revised manuscript. The DC-PTS amplitude increases with optical power, while the noise level remains nearly unchanged. Hence, the SNR increases with optical power.

4. The schematic of the profile of the HC-ARF is a bit confusing to interpret. I think this is because the rods are the same color as the surrounding hollow areas. Maybe the rods could be made with a cross-hatch pattern to distinguish them better?

Reply: The scanning electron microscope (SEM) image of the HC-ARF is added in Fig. 2 for clarification.

5. The gas inlet and outlet method was a bit confusing to me. It sounds like the SMF is spliced the HCF, but that a cap is left between the SMF and HCF. How is this achieved?

Reply: Instead of the traditional fusion splicing of optical fibers, we align the fibers manually to generate a small gap between them. Then all the fibers are fixed using UV epoxy glue to ensure stability and reliability. An inset graph of the fiber splicing has been added in Fig. 2 for clarification. Besides, we have replaced 'mechanically splicing' by 'manually aligning' in the Methods of the revised manuscript.

6. One final point is that it would be interesting to have a comparison of the SNR of the photothermal system compared to a linear absorption measurement. This can be done by estimating the SNR of the linear measurement using the published absorbance noise from other systems as well as the HITRAN calculated absorbance spectrum. Is the maximum achievable SNR (assuming higher laser power for photothermal) higher for photothermal? If not, then then what would be the expected advantage of photothermal spectroscopy?

Reply: Compared to absorption spectroscopy, PTS is an indirect measurement method with the sensitivity relevant to laser power and optical path. Thus, one advantage of PTS is to enhance the SNR by using a high-power pump source. DC-PTS also has other advantages such as small sample volume and compact configuration. It is not easy or unfair to directly compare dual-comb absorption spectroscopy, DC-PTS or DC-PAS in terms of the maximum achievable SNR. The direct linear absorption method has a long-term development, leading to many fantastic techniques for enhancing SNR such as feed-forward dual-comb spectroscopy [1] and adaptive dual-comb spectroscopy [2]. Hence, the main purpose of this paper is to report the new concept and its potential impact on spectroscopic measurements. We have discussed several possible ways to further improve its performance, which need a lot of follow-up studies to evaluate the maximum achievable SNR.

[1] Chen, Z., Hänsch, T. W., & Picqué, N. *Mid-infrared feed-forward dual-comb spectroscopy. Proc. Natl. Acad. Sci.* **116**, 3454-3459 (2019).

[2] Ideguchi, T., Poisson, A., Guelachvili, G., Picqué, N., & Hänsch, T. W. *Adaptive real-time dual-comb spectroscopy. Nat. Commun.* **5**, 3375 (2014).

REVIEWER COMMENTS

Reviewer #1 (Remarks to the Author):

The authors have answered all my questions and after the discussion, I believe that the manuscript is suitable for publication.

Reviewer #2 (Remarks to the Author):

The authors have addressed many of the questions and concerns raised by the reviewers and many technical questions I had had been answered satisfactorily. There are a few remaining issues they should further clarify hopefully, before it is accepted for publication:

As to the role and effect of FPI on dual-comb photothermal spectroscopy, which is a key contribution of this work, I still have some questions about their answers and the additional new data provided this time. I think the authors realize that the end reflection is also essential to form the FPI in their current experimental configuration. Therefore, the associated spectral modulation is unavoidable. This actually somewhat has some side effects, likely negative ones (as I'll explain below), when working with a broadband scheme like the dual-comb one. In their reply and revised manuscript, the claim that one can suppress the fiber end reflection to eliminate the FP spectral modulation is not convincing or correct.

When they proposed to increase the length of the fiber by two orders of magnitude, which would drastically reduce the spectral modulation period, it would make it rather difficult to resolve those similarly narrow absorption features, especially when they are weak. Seems that the authors didn't fully consider that factor yet.

Another issue is: in lieu of the new data shown in the supplement Fig S3, there should be noticeable amplitude modulation in the optical spectrum, thus the dual-comb spectrum as well, after the FPI. Yet the dual-comb spectrum shown in the Fig. 3b inset is rather smooth without such modulation. A good explanation is required for this discrepancy.

Reviewer #3 (Remarks to the Author):

Thanks to the authors for the edits and reply. Overall, my comments were addressed sufficiently and I recommend that paper for publication.

Manuscript number: NCOMMS-21-46571A

Title: Dual-comb Photothermal Spectroscopy

Correspondence Authors: Prof. Qiang Wang, Dr. Zhen Wang, and Prof. Wei Ren

Response to Reviewers

We appreciate the reviewer's comments and suggestions. We provide here the point-by-point response. Reviewer's comments are in black, responses are in blue.

Reviewer #1:

The authors have answered all my questions and after the discussion, I believe that the manuscript is suitable for publication.

Reply: We are happy to know that the reviewer is satisfied with our revision.

Reviewer #2:

The authors have addressed many of the questions and concerns raised by the reviewers and many technical questions I had had been answered satisfactorily. There are a few remaining issues they should further clarify hopefully, before it is accepted for publication:

As to the role and effect of FPI on dual-comb photothermal spectroscopy, which is a key contribution of this work, I still have some questions about their answers and the additional new data provided this time. I think the authors realize that the end reflection is also essential to form the FPI in their current experimental configuration. Therefore, the associated spectral modulation is unavoidable. This actually somewhat has some side effects, likely negative ones (as I'll explain below), when working with a broadband scheme like the dual-comb one. In their reply and revised manuscript, the claim that one can suppress the fiber end reflection to eliminate the FP spectral modulation is not convincing or correct.

Reply: It is admitted that, for the current experiment, the end reflections cause the spectral modulation of the pump light. However, as we discussed in the Supplementary, the amplitude variation is less than 5%. Such a small variation is acceptable in many applications. This issue can be further mitigated by applying a broadband anti-reflection (AR) coating to the fiber facets over the wavelength range of the pump comb so that a FPI is not formed for the pump. Note that such a broadband AR-coating technology is quite matured. Meanwhile, the probe wavelength can be selected beyond the spectral range of the AR coating so that the reflections for the probe are still relatively high to form a FPI with a reasonable interference fringe contrast for the detection of photothermal phase (or RI) modulation. The text on page 9 in the manuscript is revised to make this clearer.

To support our argument, we provide here some preliminary data that we obtained recently in another photothermal system as private communications. The system uses a 2.00- μm distributed-feedback laser as the pump and a 1.55 μm single-frequency laser as the probe. A single absorption line of CO_2 is exploited in a 15 cm HC-ARF-based FPI. Particularly, an AR-coating is added to achieve only 0.3% reflection for the pump wavelength and 20% reflection for the probe wavelength. As shown in the following Fig. R1, it is clear that the fringe noise (or spectral modulation) is removed by applying the AR-coating for the pump wavelength. Besides, the amplitude of photothermal signal increases due to the higher reflection for the probe wavelength with the coating. Hence, we believe the similar method can be applied to our dual-comb photothermal spectroscopy, which deserves a thorough investigation in the future.

Fig. R1. Effect of the AR-coating on the improved photothermal signal. The comparison of the photothermal measurements shows that the additional AR-coating for the pump laser near 2004 nm mitigates the spectral modulation (inset graph, basically removes the Fabry-Perot fringes). The coating has a higher reflection at the probe wavelength so that the amplitude of photothermal signal is also increased as compared to that without coating.

When they proposed to increase the length of the fiber by two orders of magnitude, which would drastically reduce the spectral modulation period, it would make it rather difficult to resolve those similarly narrow absorption features, especially when they are weak. Seems that the authors didn't fully consider that factor yet.

Reply: We have considered the fact that the spectral modulation period reduces with the increased fiber length. The spectral modulation can be mitigated using the method mentioned in the reply to the previous question. Additionally, the photothermal signal increases linearly with the fiber length while the amplitude of spectral modulation (if still exists) remains almost unchanged. Hence, we should be able to minimize the effect of the spectral modulation.

Another issue is: in lieu of the new data shown in the supplement Fig S3, there should be noticeable amplitude modulation in the optical spectrum, thus the dual-comb spectrum as well, after the FPI. Yet the dual-comb spectrum shown in the Fig. 3b

inset is rather smooth without such modulation. A good explanation is required for this discrepancy.

Reply: As stated both in the caption of Fig. 3 and the main text (paragraph above Fig. 3), the inset of Fig. 3(b) shows the dual-comb spectrum of the pump light measured by the photodetector PD1 in Fig. 2. The pump light directly impinges on PD1 without passing through the FPI, so the signal is not affected by the spectral modulation and is smoother.

Reviewer #3:

Thanks to the authors for the edits and reply. Overall, my comments were addressed sufficiently and I recommend that paper for publication.

Reply: We are happy to know that the reviewer is satisfied with our revision.

REVIEWERS' COMMENTS

Reviewer #2 (Remarks to the Author):

This version of the manuscript and the reply to my answers addressed the questions I raised last time, and I'm satisfied with them and it is OK to accept it for publication, to my opinion.

Manuscript number: NCOMMS-21-46571B

Title: Dual-comb Photothermal Spectroscopy

Correspondence Authors: Prof. Qiang Wang, Dr. Zhen Wang, and Prof. Wei Ren

Response to Reviewer #2:

This version of the manuscript and the reply to my answers addressed the questions I raised last time, and I'm satisfied with them and it is OK to accept it for publication, to my opinion.

Reply: We are happy to know that the reviewer is satisfied with our revision.